# Potential Effect of Porosity Evolution of Cemented Paste Backfill on Selective Solidification of Heavy Metal Ions

**DOI:** 10.3390/ijerph17030814

**Published:** 2020-01-28

**Authors:** Yixuan Yang, Tongqian Zhao, Huazhe Jiao, Yunfei Wang, Haiyan Li

**Affiliations:** 1Institute of Resources and Environment, Henan Polytechnic University, Jiaozuo 454000, China; yangyixuan@hpu.edu.cn (Y.Y.); zhaotq@hpu.edu.cn (T.Z.); 2International Joint Research Laboratory of Henan Province for Underground Space Development and Disaster Prevention, School of Civil Engineering, Henan Polytechnic University, Jiaozuo 454003, China; wyf_ustb@126.com; 3School of Materials Science and Engineering, Henan Polytechnic University, Jiaozuo 454003, China; lihaiyan@hpu.edu.cn

**Keywords:** cemented paste backfill, tailings management, underground water pollution, heavy metal ions, selective solidification

## Abstract

Cemented paste backfill (CPB) is a common environmentally friendly mining approach. However, it remains undetermined whether CPB pollutes underground mine water. Tank leaching analysis of a CPB mass in distilled water was performed for 120 d, and water quality was tested in situ for a long-term pollution assessment. Computerized tomography was also used to determine the CPB micro-pore structure and ion-leaching mechanism. The dissolved Zn^2+^, Pb^2+^ and As^5+^ concentrations in the leachate peaked at 0.56, 0.11 and 0.066 mg/L, respectively, whereas the Co^2+^ and Cd^2+^ concentrations were lower than the detection limit. The CPB porosity decreased from 46.07% to 40.88% by soaking, and 80% of the pore diameters were less than 13.81 μm. The permeability decreased from 0.8 to 0.5 cm/s, and the quantity, length, and diameter of the permeate channels decreased with soaking. An in-situ survey showed novel selective solidification. The Zn^2+^ concentration in the mine water was 10–20 times that of the background water, and the Pb^2+^ concentration was 2–4 times the regulated value. Although the Pb^2+^ content decreased significantly with mining depth, there remains a serious environmental risk. Mine water pollution can be reduced by adding a solidifying agent for Pb^2+^ and Zn^2+,^ during CPB preparation.

## 1. Introduction

Massive volumes of tailings and wastewater are generated into the environment during mining and mineral processing [1,2]. There is a risk of tailings causing dam failure, depending on the method used for dam construction [3,4]. On 26 January, 2019, a serious dam failure accident led to over 300 people being killed or missing, in a Vale S.A. mine in Minas Gerais, Brazil [5]. Furthermore, the filtrate and runoff from tailing ponds contain numerous heavy metal ions and toxicants, and these create a significant issue with regard to the environment and human health [6,7]. The concentration of metals and acids in water close to mines and mine wastes, is significantly high [8,9].

The cemented paste backfill (CPB) approach is a high-efficiency method for solving the environmental and geological challenges induced by transporting tailings underground [10,11,12]. CPB has been widely used in tailing recycling [13,14], underground stope supporting [15] and open pit stabilising [16,17]. However, tailing particles and processing water contain numerous types of heavy metals. Most heavy metal ions can be recovered via thickening the water that is recycling back to the processing plant, but some are still delivered underground in the CPB mass [18]. Initially, in the thickener, a portion of the heavy metal ions and flotation reagents adsorbs to the flocculants [19], which combine with the tailing particles to form large aggregates. These aggregates settle to the bottom of thickener. The discharged thickener underflow is then pumped underground as CPB material. Second, the toxicants in the CPB material are fixed in the concreted mass underground, thereby avoiding or reducing diffusion [20]. The concreted CPB mass is a porous material, with cracks and pores [21,22,23]. 

Heavy metal solidification and leaching from cement-based materials have gained significant research attention [24,25]. In terms of solidification, Cu and Pb are predominantly absorbed by calcium silicate hydrate gels (C–S–H), while Cd^2+^, Ni^2+^ and Zn^2+^ mainly precipitate as hydroxides within the intergranular pores [26]. The mobility of heavy metals in the Solidification/Stabilization system (S/S system), follows the order of Cd^2+^ > Pb^2+^ > Zn^2+^ > Cu^2+^ > Ni^2+^ [27]. Geopolymers have shown a high degree of solidification of Pb^2+^, Cd^2+^, Mn^2+^ and Cr^6+^ in fly ash [13,28]. The ionic radius of Pb^2+^ is only approximately 13% larger than that of Ca^2+^, therefore, some authors have suggested replacing Ca^2+^ with Pb^2+^ [11].

In terms of leaching, the effectiveness in reducing the leachability of Ba^2+^, Cd^2+^, Cu^2+^, Ni^2+^ and Pb^2+^ was shown to be easy for natural zeolite-blended cement pastes [29]. Metals in cement were shown to be leachable in various media, in the descending order of Cu^2+^, Cd^2+^, Pb^2+^, Zn^2+^, Mn^2+^ and Sb^3^, but were not leached in simulated seawater, groundwater or acid rain [30]. The microstructure of cementation wastes containing Pb^2+^, Cd^2+^, As^5+^ and Cr^6+^ has been investigated [9,31,32]. Metal leaching in the pH range of 6–8, was shown to decrease in the following order: Cr^6+^ > Cd^2+^ > Pb^2+^ > As^5+^ [9,33]. High concentrations of Zn^2+^ and Ni^2+^ were observed in water and sediment soils, in streams located near the Chambishi copper mine in Zambia [32], however, no studies have yet determined whether CPB causes long-term pollution to underground water.

The structure of the present study was as follows: First, in a period of months (short term), the solidification effect of the CPB mass for specific metal ions was evaluated by a 120-day indoor soaking test. Second, over a period of years (long-term), the quality of the underground mine water was investigated, in terms of long-term solidification. Third, the physicochemical adsorption and leaching seepage mechanics were explained, via a leaching-out assessment. The influence of the real pore structure and fractures of the CPB mass on permeability, was studied by computed tomography (CT), nuclear magnetic resonance (NMR) and visual analysis. Finally, the question of whether the CPB mass pollutes underground mine water, was answered.

## 2. Materials and Methods

### 2.1. Short-Term Leaching Test

The EA NEN 2004 tank test [33] (Figure 1) was adopted to assess the leaching characteristics of metal ions from porous monoliths, which involved the leaching of cement mortar specimens, with distilled water as the leachate. The leachates were tested at six specific periods (10, 30, 50, 70, 90 and 120 d). The CPB blocks were prepared with a copper mine tailing and Portland cement, and the tailings-to-cement ratio was 12:1 by weight, the same as the underground CPB. The block size was 70.7 mm × 70.7 mm × 70.7 mm. The leachates were detected by a water quality heavy metal analyzer, to determine to ion concentration, as shown in Figure 1d. 

### 2.2. Long-Term Solidification Performance Evaluation

Underground mine water was sampled to evaluate leaching over years, as a long-term period. The CPB operation was established in 2013, in the Chambishi copper mine in Zambia, as shown in Figure 2. The tailing slurry was treated with a deep cone thickener, to output a high-concentration underflow; the paste underflow was then mixed with cement and transported underground through a pipeline, as shown in Figure 2c. The cement-sand ratio was 1:12 to 1:30, and the target uniaxial compressive strength (UCS) was 0.3–0.5 MPa. The CPB mass has been soaking in the mine water more than 6 years.

As shown in Figure 3, the underground mine water was sampled at 3 locations: –116, –200, and –264 m. It must be noted that the water was sampled directly from the roof, and the samples were not mixed with the mining treatment water.

### 2.3. Permeability and Porosity of the CPB

The control CPB samples were tested to determine the permeability performance in the same period as indicated above. To determine the CPB porosity, the water in the pores was analyzed by NMR spectroscopy. A high-precision industrial micro-CT scanning system was adopted for porosity and pore visualization [34,35]. The reconstructed pore models were analyzed, to explain the heavy metal ion leaching process, as shown in Figure 4 and Figure 5.

### 2.4. Tailings 

The contents of five toxicant heavy metal ions in the tailings used in this study, are shown in Table 1. The main ion is Co^2+,^ with a content of 390.26 mg/kg. The contents of the other 4 metals, Zn^2+^, As^5+^, Cd^2+^ and Pb^2+^, were approximately 70 mg/kg. 

## 3. Results and Discussion

### 3.1. Ion Leaching from the CPB in the Short-Term

The ion concentrations in the leachate of Co^2+^, Zn^2+^, Pb^2+^, As^5+^ and Cd^2+^ are discussed in this section. The Co^2+^ and Cd^2+^ concentrations were lower than the detection limit. 

#### 3.1.1. Dissolved Zn^2+^ in Leachate

The Zn^2+^ content was higher than that of the other metals in the leachate, as shown in Figure 6. The content increased from 0.05 to 0.56 mg/L during the first 70 d, indicating that the Zn^2+^ sealed in the CPB blocks, had diffused into the water. In the following 70–120 d, the value decreased to 0.49 mg/L, demonstrating that the dissolved ions in the water were re-adsorbed into the CPB blocks, after a critical soaking period.

Zn^2+^ ions will react with OH^−^ to generate an amorphous granular layer of Zn(OH)_2,_ that can attach to the surface of hydrated cement particles. The amorphous granular material is a mixture of zinc silicate gel and poorly crystallised C–S–H gel, that is unstable, and readily releases zinc ions during leaching—the adsorption is unstable and reversible, which is the reason for the increased dissolved Zn^2+^ concentration. In contrast, in the cement hydration process, Zn^2+^ will partially substitute and react with Ca^2+^ in the C–S–H gel to form Ca[Zn(OH)_3_·H_2_O]_3_, a crystalline hydrate of calcium and zinc. The reaction product is chemically stable, and thus the Zn^2+^ ions cannot be released from the crystal structure. This explains the lower concentration in the leachate than the original value in the tailings. 

#### 3.1.2. Dissolved Pb^2+^ in Leachate 

As shown in Figure 7, the Pb^2+^ content was lower than that of Zn^2+^ in the first 70 days, increased from 0.02 to 0.11 mg/L, then decreased to 0.08 mg/L after 120 days. These drops were caused by the re-adsorption of the CPB. The main cement hydration products of the CPB were C–S–H gel, Ca(OH)_2_, *AFt*, and *AFm*. Among these, C–S–H gel has an extremely high specific energy and ion exchange capacity; ions can be fixed by adsorption, symbiosis and interlayer position replacement, etc. Ettringite can seal several kinds of ions in its crystal columns and channels by chemical substitution.

Pb^2+^ can replace Ca^2+^ in the C–S–H gel, by entering the C–S–H gel structure and bonding with Ca^2+^ and Si^4+^. Pb^2+^ changes the C–S–H gel density and nanostructure [22], and a large number of acicular crystals increase the porosity of the CPB mass. This contributes to the infiltration and physical adsorption of Pb^2+^ ions, however, the structural change also increases the crack and fracture quantity and width, as well as the loose material. This physical adsorption can be defined as weak in strength. The fractures create seepage flow-channels in the CPB mass that increase the contact between adsorbed ions and the leaching solution. Consequently, previous studies [3,28] have shown that cement can solidify Pb^2+^ ions, and the solidification effect of the CPB mass on Pb^2+^ is lower than expected. 

#### 3.1.3. Dissolved Cd^2+^ in Leachate

The electric potential of Cd^2+^ is similar to that of Ca^2+^, as the difference in their ionic radii is less than 15%, therefore, Cd^2+^ can replace Ca^2+^ in the cement hydration product. The replacement process forms the corresponding Cd silicate crystal gel, but the hydration product structure does not undergo a crystal lattice distortion. Cd^2+^ is captured by the C–S–H in two ways: the amorphous hydrated calcium silicate formed in the precipitation method, and ion exchange with the C–S–H [25]. After ion exchange, approximately 30% of Cd^2+^ can adsorb into the C–S–H in both free and solidified forms. The captured Cd^2+^ could originate not only from enrichment of co-precipitation, but also as a hydration product of both combinations, as shown in Figure 8. Rather than physical adsorption, chemical consolidation played a major role in the Cd^2+^ solidification mechanism.

#### 3.1.4. Dissolved As^5+^ and Co^2+^ in Leachate

As shown in Figure 9a, the ions in the CPB blocks diffused into the leachate. The As^5+^ concentration increased from 0.053 to 0.066 mg/L after 30 d of soaking. Beyond 30 d, the concentration curve decreased rapidly, owing to the re-adsorption process. 

The C–S–H gel is a layered silicate, and thus As^5+^ more readily consolidates in an ettringite lattice structure. The majority of the captured As^5+^ forms arsenate [28] that physically attaches or adsorbs onto the surface of the hydration product, as shown in Figure 9b. 

The content of Co^2+^ in the leachate was extremely low. The particle surface bond energy is reduced by the destruction of Ca–O bonds during cement grinding. Co^2+^ can replace Ca^2+^ in Ca–O bonds on the cement particle surfaces and inside the hydration product. Furthermore, Co^2+^ does not change the crystal lattice structure of the hydration product. 

### 3.2. Long-Term Leaching Performance

#### Mine Location

The metal ion concentrations in the underground mine water are shown in Table 2. In the north-western part of Zambia, the concentrations of the target heavy metals were higher than the corresponding international standards [36]. The concentrations of Co^2+^, Ca^2+^, and As^5+^ in the mine water were lower than the experimental and regulated values [37,38]. The CPB body has a considerable consolidation effect on the target metal ions. However, the Zn^2+^ and Pb^2+^ results shown in Table 2 require further study.

The concentration of Zn^2+^ was also lower than the regulated value and decreased with mining depth. At a depth of –116 m, the Zn^2+^ concentration was 0.49 mg/L, which decreased to 0.26 mg/L at –264 m. However, the concentration of Zn^2+^ in the mine water was still 10–20 times more than that of the background water, therefore, in the natural environment, Zn^2+^ poses a major challenge to the prevention and control of water pollution.

The concentration of Pb^2+^ in the mine water was 0.236 mg/L at a depth of –116 m, which decreased to 0.114 mg/L as the depth decreased to –264 m. The Pb^2+^ concentration was 2–4 times more than the regulated value and 100–500 times more than that of the background water, however, despite the Pb^2+^ concentration decreasing significantly with mining depth, there remains a significant environmental containment risk. Considering their amphoteric nature, Pb^2+^-containing hydroxides encounter high-concentrations of OH^−^ and generate soluble Pb(OH)^3−^, which accounts for the dissolution and re-precipitation of lead salts throughout the cement hydration process. 

With regard to cement-based concrete and contaminated soil treatment, cement as a common heavy metal solidification reagent has been widely examined [39,40,41,42]. However, different from the materials in previous reports, CPB has a very low ratio of cementitious materials; the lattice structure produced by cement hydration only has an acceptable adsorption effect on a few specific ions and cannot effectively solidify Pb or Zn ions. Therefore, a solidifying agent for heavy metal ions should be added for future CPB operation.

## 4. Discussion

### 4.1. Porosity and Permeability of CPB

#### 4.1.1. Porosity

The NMR results showed that the porosity decreased from 46.07% to 40.88% over the course of 120 d, as shown in Figure 10. The cement hydration process continued during soaking, and the new hydration product made the solidified matrix structure become denser. The hydration process increased the diffusion resistance. 

Based on the pore size distribution, 50% of the pores were smaller than 5.6 μm and 80% were smaller than 13.81 μm. The pores with diameters less than 5 μm, were mostly isolated and not considerably affected by the concrete permeability performance. In contrast, the pores larger than 10 μm were connected and thus had a significant influence on CPB permeability. The total porosity reached 100.3% of the final porosity at 90 d, when the hydration process was complete.

#### 4.1.2. Permeability

The permeability test showed that the permeability was proportional to the porosity. The permeability decreased from 0.8 to 0.5 cm/s with soaking time, as the continuous hydration reaction blocked the seepage flow channels. The finer particles separated from the matrix under long-term soaking, which blocked the channels and increased the seepage resistance. The quantity, length, and diameter of the seepage channels decreased with soaking time. As shown in Figure 11, the quantity of main channels in the 1000 μm × 1000 μm × 1000 μm cubic CPB blocks decreased from 10 to 4 over 120 d, a drop of 60%. The average streamlined length decreased from 0.8 to 0.3 mm, a drop of 62.5%, and the average channel diameter decreased from 1.1 to 0.5 μm, a drop of 54.5%. 

The simulation results showed that the permeability decreased from 1.2 to 0.7 cm/s, with immersion time. The hydraulic conductivity determined experimentally and via simulation were of the same order of magnitude; the difference was caused by the insufficient resolution of CT imaging. Parts of the narrow-hole channels which were less than the detection limit were ignored.

It is difficult to discover seepage blind zones through macroscopic seepage experiments. Such zones significantly reduce the hydraulic conductivity, in which the geometrical characteristics of the pore network play an important role. Once the dominant flow is formed in the CPB, a seepage blind zone is inevitable. The solution cannot flow through these areas, and the solute transfer process can only occur by diffusion, resulting in low ion leaching rates around these areas. 

#### 4.1.3. Solidification Mechanism

Heavy metal ions are solidified into the cement hydration product by physical solidification, substitution, precipitation and isomorphic substitution, which prevent migration and diffusion into the environment, and thereby reduce the toxicity of the heavy metals. The C–S–H can hold or adsorb metal ions by ion replacement and surface electronegativity during the hydration process. When the radius of heavy metal ions is close to that of the ions in ettringite, Al^3+^, Ca^2+^ and SO_4_^2−^ can be replaced by the corresponding heavy metal ions, to form ettringite crystals. 

As shown in Figure 12, the major product of C–S–H was a mixture of poorly crystallised particles, with different morphologies. It has been reported that there are four types of C–S–H defined according to morphology [43,44]: (1) fibrous (acicular crystal), (2) columnar reticular network [45], (3) equate grain morphology [46], and (4) inner product morphology. It is generally believed that the adsorption capacity of cement minerals to waste is ranked as follows: C3A>C3S>C4AF>C2S. Heavy metal ions can also be adsorbed in other ways. An alkaline environment can cause heavy metal ions to form hydroxide precipitates, and physical encapsulation and chemical binding to ettringite can occur. The easier way is to adsorb or settle on the surface of tailing particles. 

## 5. Conclusions

Different from the traditional results of cement-based concrete materials analysis, the CPB material showed a significant selectivity in the adsorption of heavy metal ions. In the short term, CPB can adsorb Co^2+^, Cd^2+^ and As^5+^ effectively. Long-term sampling results showed that the contents of Co^2+^, Cd^2+^ and As^5+^ in mine water were lower than the experimental and regulated values. Within 120 days of CPB leaching, the Co^2+^ and Cd^2+^ concentrations were lower than the detection limit, whereas the Zn^2+^, Pb^2+^ and As^5+^ concentrations reached 0.49, 0.08, and 0.066 mg/L, respectively.

In the long term, a large amount of Zn^2+^ and Pb^2+^ leached from the CPB materials. The content of Zn^2+^ was 10–20 times that of the background water, after eight years of immersion and leaching of the CPB mass. Most seriously, the Pb^2+^ concentration in the mine water was 0.114 mg/L, which is 2–4 times the regulated value and 100–500 times that of the background water.

CPB technology has been considered attractive, but until now, it had not been well investigated whether it pollutes underground mine water. The results of this study indicate that controlling underground mine water pollution by the CPB mass should receive more attention, especially the possibility of adding a heavy metal ion solidification agent.

## Figures and Tables

**Figure 1 ijerph-17-00814-f001:**
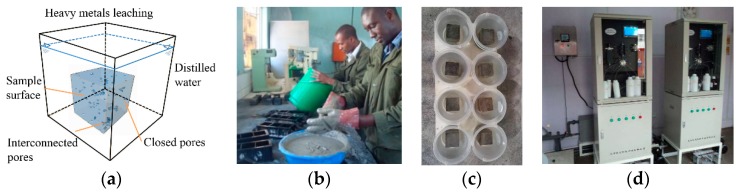
Cemented paste backfill (CPB) block leaching and leachate ion testing. (**a**) EA NEN 2004 tank test scheme. (**b**) CPB block preparation. (**c**) Blocks leaching. (**d**) Water quality heavy metal analyzer.

**Figure 2 ijerph-17-00814-f002:**
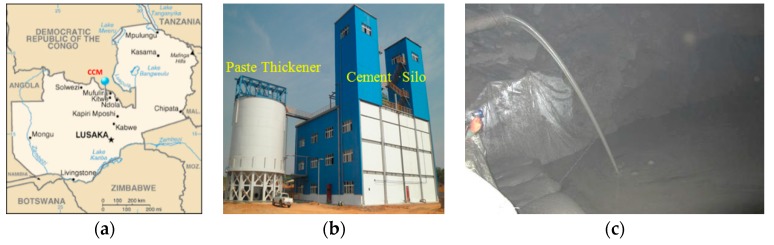
(**a**) Location of the Chambishi copper mine. Photographs of the (**b**) CPB plant and (**c**) underground stope backfill.

**Figure 3 ijerph-17-00814-f003:**
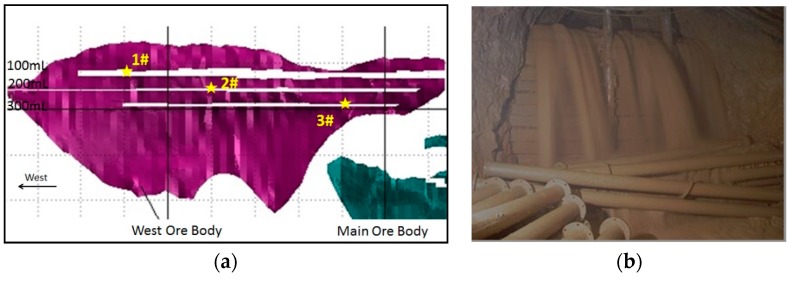
(**a**) Underground mine water sampling locations. (**b**) Photograph of the mine water from the roof.

**Figure 4 ijerph-17-00814-f004:**
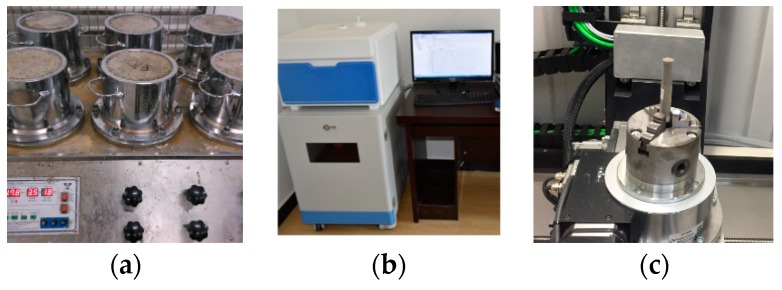
CPB block permeability and porosity testing. Equipment for (**a**) permeability, (**b**) NMR, and (**c**) CT analyses.

**Figure 5 ijerph-17-00814-f005:**
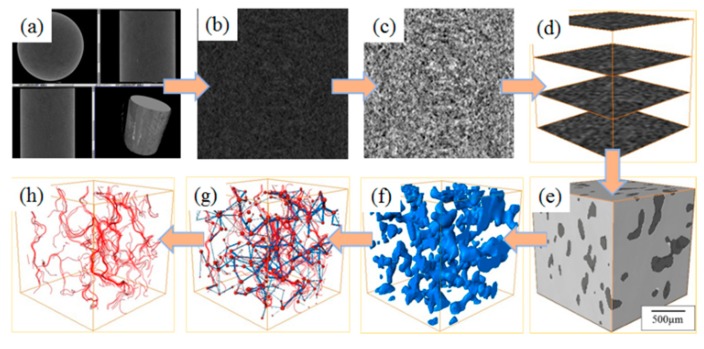
CPB block three-dimensional reconstruction and seepage simulation. (**a**) CT result, (**b**) CT image, (**c**) binary image, (**d**) images reconstruction, (**e**) 3D reconstruction model, (**f**) pore recognition, (**g**) pore network model, (**h**) pore seepage simulation.

**Figure 6 ijerph-17-00814-f006:**
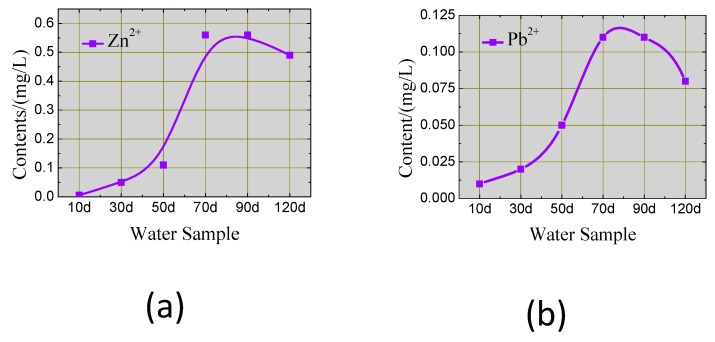
Dissolved (**a**) Zn^2+^ and (**b**) Pb^2+^ concentrations in the leachate.

**Figure 7 ijerph-17-00814-f007:**
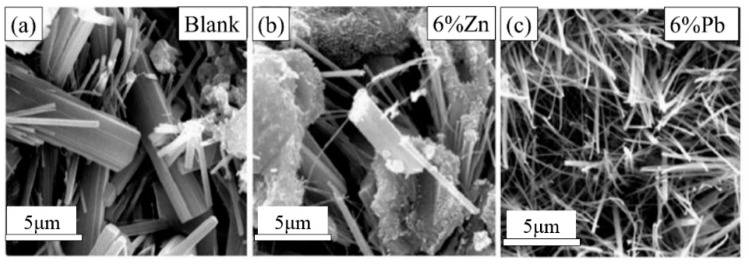
Change in shape of C–S–H induced by Zn^2+^ and Pb^2+^.

**Figure 8 ijerph-17-00814-f008:**
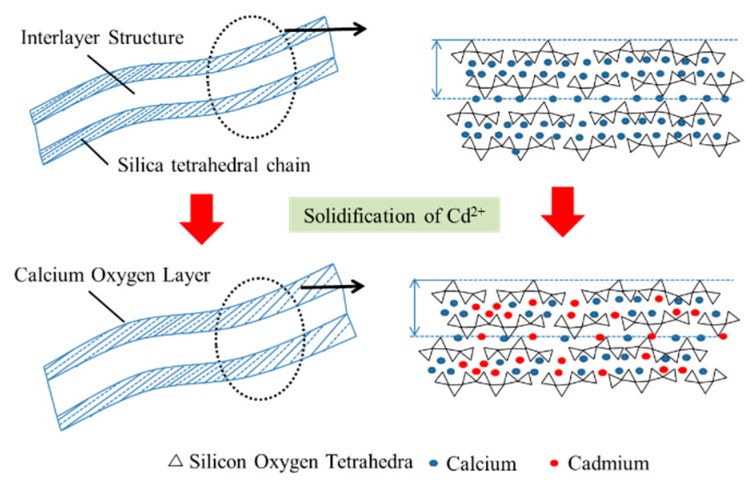
Impact of Cd^2+^ on the C–S–H gel structure.

**Figure 9 ijerph-17-00814-f009:**
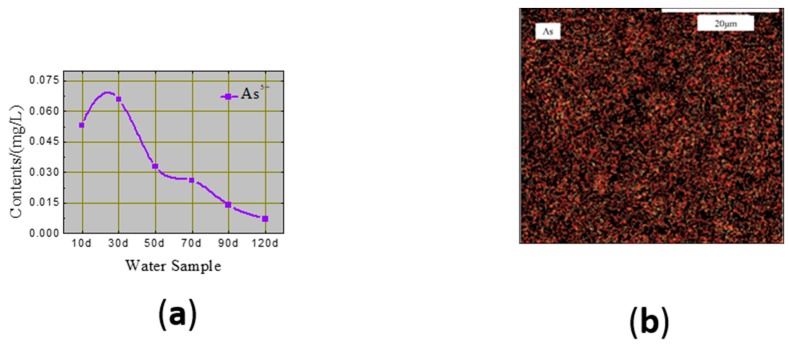
(**a**) As^5+^ concentration in leachate and (**b**) corresponding micro-morphology, determined by X-ray imaging.

**Figure 10 ijerph-17-00814-f010:**
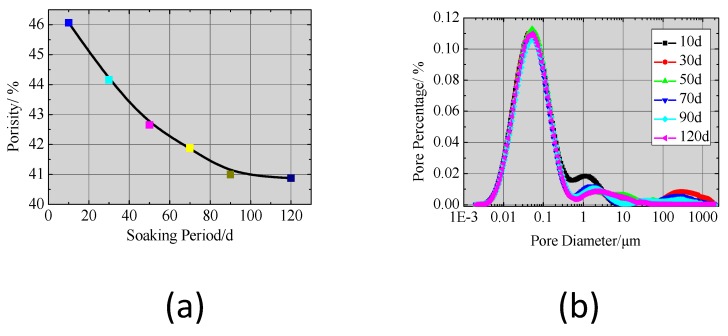
CPB mass (**a**) porosity vs. soaking period and (**b**) pore size distribution.

**Figure 11 ijerph-17-00814-f011:**
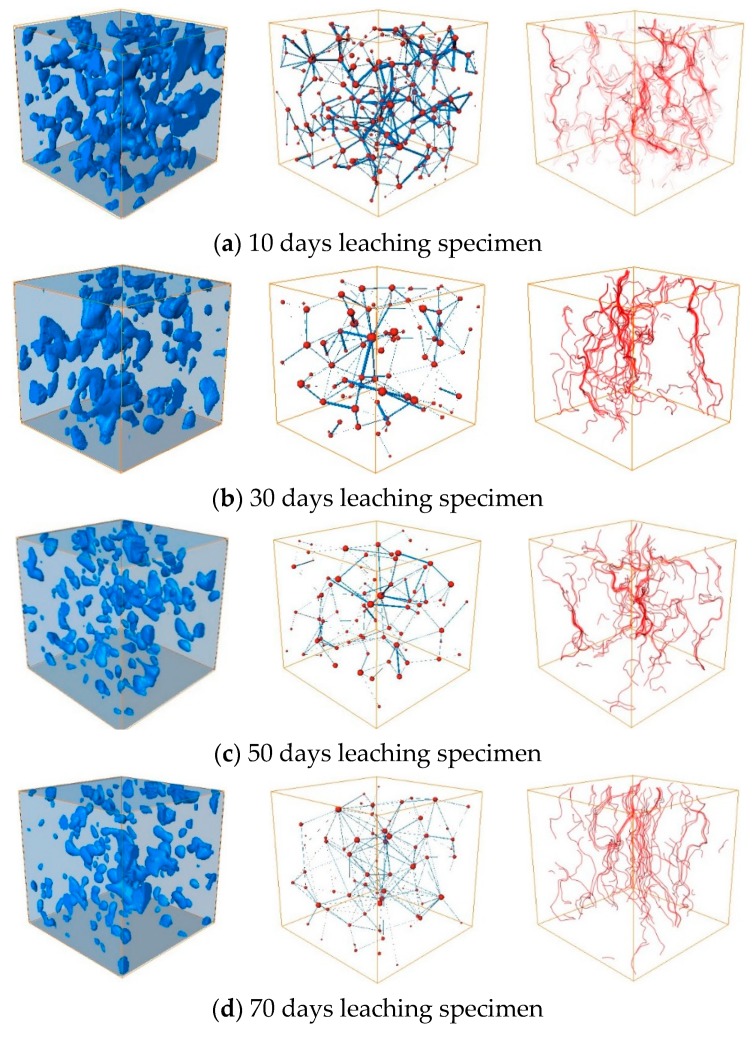
Three-dimensionally reconstructed visualised pores, pore network models, and seepage flow channel simulations.

**Figure 12 ijerph-17-00814-f012:**
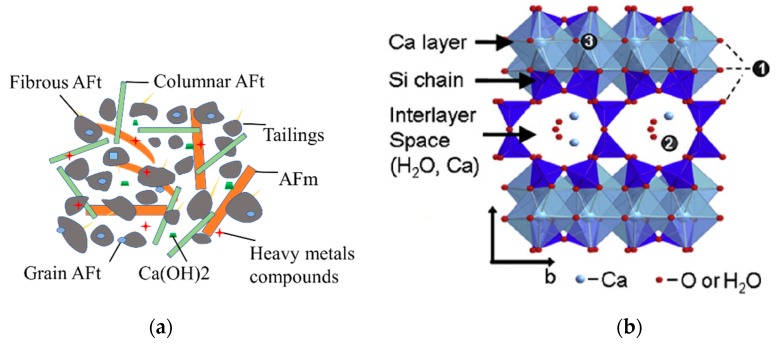
(**a**) Macro- and (**b**) molecular-scale illustrations of cement hydration adsorption mechanism for heavy metal ions.

**Table 1 ijerph-17-00814-t001:** The contents of the main toxicant heavy metals in the tailings (units: mg/kg).

Component	Co^2+^	Zn^2+^	Pb^2+^	As^5+^	Cd^2+^
Content	390.26	68.30	72.31	70.70	69.63

**Table 2 ijerph-17-00814-t002:** Heavy metal ion concentrations in the underground mine water (mg/L).

Sample	Co^2+^	Zn^2+^	Pb^2+^	As^5+^	Cd^2+^
−116 m 1# mine water	0.0082	0.49	0.236	0.049	<0.001
−200 m 2# mine water	0.0078	0.36	0.168	0.045	<0.001
−264 m 3# mine water	0.0013	0.26	0.114	0.038	<0.001
Regulated value [Nachiyunde et al., 2013; 2013a]	0.01	5	0.05	0.058	0.005
Background water [Psutka et al., 2011]	0.0066	0.0239	0.00042	0.58	0.00159

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
