# Peer review of "Potential Effect of Porosity Evolution of Cemented Paste Backfill on Selective Solidification of Heavy Metal Ions"

_ijerph, 2020, doi:10.3390/ijerph17030814_

Round 1

Reviewer 1 Report

Comments on manuscript ijerph-672405-peer-review-v1 entitled ‘The Potential Effect of Porosity Evolution of CPB on Selective Solidification Performance of Heavy Metals Ions’ authored by Yang et al., submitted to International Journal of Environmental Research and Public Health (IJEPH).

This study is not well presented, leaving it a lot of unclearness in methodology and result description. Although substantial work was done by this study, the manuscript does not provide a picture of what has been achieved and how the results come out from the experiments. It is hard to follow the flow of the manuscript. Also, there seems lack of connection between the experiments and case study. Thus, a major revision is required to improve the presentation of this study.

Few minors need the authors’ attention that scale bar is required for photos (b) and (c) of Figure 3. The caption of Figure 5 needs revised to reflect the context.

The manuscript needs reformatted to be consistent with the requirement of IJEPH. Many queries or comments are marked on the edited PDF manuscript for consideration of the author when a revision is made.

Author Response

Thanks a lot for your comments, I aggree all the points you figured out. I have addressed these questions in the revised manuscript, re-structured the outline. The language has been improved by extensive editing. 

Comments 1

This study is not well presented, leaving it a lot of unclearness in methodology and result description. Although substantial work was done by this study, the manuscript does not provide a picture of what has been achieved and how the results come out from the experiments. It is hard to follow the flow of the manuscript.

Answers 1

In methodology and result description, the process of specimen preparation, leaching test approach, CT scanning and reconstruction, water quality test has been added into the revised manuscript. The blocks preparation, quantity and dimension has been explained, the leaching processing and quantity has been demonstrated, the method and device of the heavy metal ions test has been described, and also, the question of three-dimension reconstruction of pore network model has been answered. The details can be found in section of Methods and Materials.

Comments 2

Also, there seems lack of connection between the experiments and case study. Thus, a major revision is required to improve the presentation of this study.

Answers 2

The presentation and structure of this manuscript has been major revised. The relationship between the experiments and the case study has been explained in section of Instruction, Methods and Materials. The experiment conducted in the study is to test the shot term solidification performance of CPB mass. The case study result is to demonstrate the long-term leaching capability of ions from CPB mass. They are addressed combine to answer the same question in two sides.

As shown in section Methods and Materials.

Comments 3

Few minors need the authors’ attention that scale bar is required for photos (b) and (c) of Figure 3. The caption of Figure 5 needs revised to reflect the context.

Answers 3

The comments have been addressed in the manuscript.

Comments 4

The manuscript needs reformatted to be consistent with the requirement of IJEPH. Many queries or comments are marked on the edited PDF manuscript for consideration of the author when a revision is made.

Answers 4

The revised manuscript has been re-structured according the Guideline. The presentation of references and sub title of figures have been revised for the IJEPH format.

Reviewer 2 Report

There many inaccuracies in the manuscript.

Abstract - "pollute groundwater" If the paste is used to backfill, eventually it will pollute the mine water and not so the groundwater resources.

Introduction

"are discharged", it may be not so. I suggest changing discharged by generated. "could cause disasters" - there is a risk of collapsing that will be variable according to the method used in the dam construction, among other things. The third sentence in snot finished. On the other hand, infiltration from tailings ponds contains... ...heavy metals ions, reagents, chemicals, acid concentration is "acidity" The second paragraph should be rewritten: "deliver the tailings ... The last sentence of the fourth paragraph should be rewritten or removed. There are many good examples of tailing dams that are quite in good shape and relatively safe and not contaminating the environment, so generalizing this, is not accurate.

Materials and methods - there is no indication of how many samples were studied, or how they were collected/prepared.

The name of the subsections shouldn't start with the ions (Zn2+).

last sentence of the 3.11 section - "That is the reason..."

Case study - last sentence of the first paragraph - Fig. 9 doesn´t show environmental pressure. - rewrite the sentence.

According to SI units system, space should be placed between the numerical value and the respective unit.

Author Response

Thanks very much for your attention on this manuscript, I aggree with your opinion in the revision letter.   I have addressed these questions and improved the language. 

Comments 1

There many inaccuracies in the manuscript.

Abstract - "pollute groundwater" If the paste is used to backfill, eventually it will pollute the mine water and not so the groundwater resources.

Answers 2

The keywords of reviewer figured out and the same problems have been revised, this study is now focus on the mine water pollution research.

Comments 2

Introduction

"are discharged", it may be not so. I suggest changing discharged by generated. "could cause disasters" - there is a risk of collapsing that will be variable according to the method used in the dam construction, among other things. The third sentence in snot finished. On the other hand, infiltration from tailings ponds contains... ...heavy metals ions, reagents, chemicals, acid concentration is "acidity" The second paragraph should be rewritten: "deliver the tailings ... The last sentence of the fourth paragraph should be rewritten or removed. There are many good examples of tailing dams that are quite in good shape and relatively safe and not contaminating the environment, so generalizing this, is not accurate.

Answers 2

The key words have been replaced, the describe of dam failure has been changed as a risk, and the related problem figured out by the reviewer have been revised.

Comments 3

Materials and methods - there is no indication of how many samples were studied, or how they were collected/prepared.

Answers 3

The samples quantity, preparation process explanation has been added in the revised manuscript, which can be proofed by the new figures.

Comments 4

The name of the subsections shouldn't start with the ions (Zn2+).

last sentence of the 3.11 section - "That is the reason..."

Answers 4

The names of the related subsections have been revised. The sentence has been revised.

Comments 5

Case study - last sentence of the first paragraph - Fig. 9 doesn´t show environmental pressure. - rewrite the sentence.

Answers 5

The sentence has been rewritten.

Comments 6

According to SI units system, space should be placed between the numerical value and the respective unit.

Answers 6

The unit of the full text has been modified according to SI units system.

Round 2

Reviewer 1 Report

The revised version of this paper has been much improved. The issues raised in the original manuscript have been resolved. This study is strongly recommended for publication of the journal.